# ROADSIGHT: A NOVEL DATASET FOR REAL-TIME INTERSECTION DETECTION IN AERIAL SCENES UNDER SEASONAL VARIATION

## ABSTRACT

Road intersections serve as critical nodes in transportation networks, and their accurate detection from aerial imagery is helpful for applications such as autonomous navigation, urban planning, and traffic management. However, existing datasets for object detection in aerial views often lack specificity to intersections, diversity in seasonal conditions, and customized annotations for unmanned aerial vehicle (UAV) captured data, which leads to challenges in model robustness and real-time performance on edge devices. This work introduces a novel dataset named ROADSIGHT (Road-Oriented Aerial DataSet for Intersection detection on edGe Hardware plaTform), a UAV-captured dataset of high-resolution RGB images collected in both summer and winter conditions, with expert bounding-box annotations for two classes: roundabouts and intersections (encompassing 3-leg T/Y and 4-leg types). We benchmark state-of-the-art models(YOLOv8, YOLOv11, YOLOv12, and RT-DETR) and identify YOLOv11s as edge suitable. Robustness is validated through cross-validation with stable mAP. This work contributes a focused, seasonally varying benchmark and corresponding baseline results for real-time intersection detection on resource constrained UAV platforms, while clearly acknowledging current limitations in geographic coverage, adverse weather, and class granularity for future extensions.

## 1 INTRODUCTION

Road intersection detection from aerial imagery is emerging in modern geospatial applications, including intelligent transportation systems (ITS), disaster response, and urban infrastructure monitoring, where precise identification can improve safety and efficiency Afrin et al. (2024). Intersections are particularly safety-critical: over 50% of severe crashes occur at or near intersections, making precise detection essential for improving traffic safety and navigation Zong et al. (2022). To address these safety challenges, intelligent systems that can identify intersections, assess their conditions, and support automated decision-making for traffic management and navigation are needed. The development of such systems requires robust computer vision models capable of accurate real-time detection across diverse environmental conditions, including seasonal variations, different lighting conditions, and weather changes that significantly affect intersection visibility and characteristics Kinnari et al. (2022). The proliferation of Unmanned Aerial Vehicles (UAVs) has enabled cost-effective data collection at high resolutions, making them ideal platforms for gathering the comprehensive datasets needed to train these robust models, yet challenges persist in handling variations such as lighting, weather, and seasonal changes that affect image quality and model generalization Hu et al. (2025). This work uses RGB imagery from standard UAV cameras without additional sensors like LiDAR, focusing on object detection frameworks suitable for real-time processing.

The motivation stems from the scarcity of specialized datasets for road intersections in UAV imagery, hindering the development of robust deep learning models for real-world scenarios (Cui et al., 2025). Difficulties include manual annotation of diverse intersection types across seasons, computational constraints for edge inference, and the need for high accuracy in complex terrains. The significance lies in paving the way for terrain-aware mapping systems that could integrate intersection detection with 3D modeling, thereby reducing reliance on expensive surveying methods. Prior efforts have not fully addressed these challenges due to limitations in dataset scale, environmental diversity, and

their predominant focus on general roads or vehicles rather than intersections (Obaid et al., 2025; Ye et al., 2025). The goal of this work is to create a multi-seasonal UAV intersection dataset and evaluate edge-optimized object detection models for real-time deployment, as shown in Figure 1.

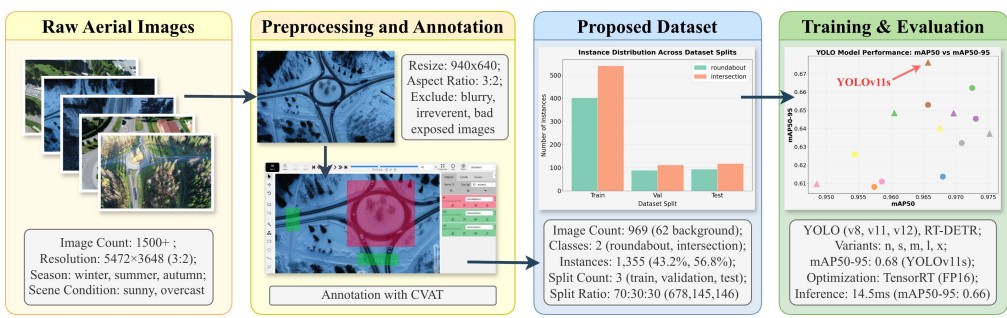

Figure 1: ROADSIGHT dataset approach overview: addressing intersection detection challenges through systematic dataset creation and optimized model evaluation for real-time UAV deployment.

Therefore, we propose a novel dataset of UAV-captured images annotated for road intersections, collected from localized test areas in Finland, under winter and summer conditions, to capture seasonal variations. The methodology involves data collection, preprocessing, training, and evaluation using different YOLO (You Only Look Once) (Redmon et al., 2016) as well as RT-DETR (Real-Time Detection Transformer) (Lv et al., 2023) models, followed by model optimization for edge deployment Sharma et al. (2025). This method emphasizes advantages such as improved mean average precision (mAP) in varying conditions, reduced inference time on edge devices with limited computational capability for enhanced geo-spatial analysis.

## 2 RELATED WORK

Datasets and methods for object detection in aerial imagery, particularly those relevant to road infrastructure and UAV-based applications, are reviewed to contextualize our novel dataset and its evaluation. They are grouped into general aerial object detection datasets and specific road-related datasets, highlighting their contributions, advantages, disadvantages, and differences from our approach. General aerial object detection datasets provide benchmarks for tasks like vehicle or small object identification but often lack targeted annotations for road intersections. For instance, the DOTA dataset comprises 2,806 high-resolution aerial images with 188,282 instances across 15 categories, including vehicles and bridges and roundabouts (Xia et al., 2018). Its advantage is the large scale and multi-sensor sourcing, supporting advanced detectors like rotated CNNs; however, its disadvantages by not including intersection-specific classes and limited environmental diversity, leading to poor generalization in seasonal variations. Similarly, the VisDrone dataset offers 10,209 images and 263 videos from drones, with 2.5 million annotations for detection and tracking, excelling in real-world drone scenarios with advantages in video sequences for dynamic analysis (Zhu et al., 2022). Yet, it focuses on pedestrians and vehicles, neglecting intersections, and suffers from occlusion issues in dense scenes. The iSAID dataset extends DOTA with instance segmentation, annotating 655,451 objects in 2,806 images across 15 categories, benefiting precise boundary delineation but limited by static satellite origins, lacking UAV-specific perspectives and seasonal data (Zamir et al., 2019). In contrast, our dataset uniquely targets road intersections with UAV-captured RGB images in winter and summer conditions, providing annotations for specific types (roundabouts or intersections), which enhances novelty in environmental robustness and integration with real-time YOLO models, unlike the broader but less specialized focus of these works.

Specific road-related datasets address infrastructure elements but vary in scope and platform. The UAVDT dataset includes 100 video clips with over 80,000 frames for vehicle detection and tracking from UAVs, advantageous for motion analysis in traffic scenarios but disadvantaged by its emphasis on vehicles rather than static intersections, with limited annotation diversity Du et al. (2018). The Roundabout Aerial Images dataset by Puertas et al. (2022) features 15,474 images from Spanish roundabouts with 985,260 instances, strong for traffic density studies in circular intersections due to its open-access nature and high instance count; however, it is restricted to roundabouts, ignor-

ing other intersection types, and lacks seasonal variations. Another work on satellite-based road intersection detection uses 14,692 images to train models for route planning, benefiting urban applications with advantages in scalability from satellite data but limited by lower resolution and absence of UAV angles or environmental adaptations Eltaher et al. (2022). In contrast, our method differentiates by combining multi-season UAV imagery with intersection-specific annotations, evaluated for edge-optimized performance, offering superior adaptability and specificity not found in these vehicle- or satellite-centric datasets.

In addition to real-world UAV datasets, several synthetic datasets have emerged that provide annotated data for aerial scene understanding. Syndrone (Rizzoli et al., 2023), MidAir (Fonder & Van Droogenbroeck, 2019), and DDOS (Kolbeinsson & Mikolajczyk, 2024) use simulation environments to generate large-scale synthetic imagery with annotations for tasks such as depth estimation, obstacle avoidance, and semantic segmentation. These datasets allow controlled environments but lack real-world complexities such as sensor noise, geographic diversity, and fine-grained intersection annotations, and are not designed for edge deployment. ROADSIGHT, in contrast, captures genuine seasonal variation (e.g., snow-covered and sunlit roads), focuses explicitly on road intersections and benchmarks real-time performance on edge hardware. This practical emphasis on real data fidelity and deployment constraints establishes ROADSIGHT as a complementary and novel benchmark for UAV-based intersection detection.

## 3 DATASET CONSTRUCTION

This section describes the construction of the ROADSIGHT dataset, including UAV-based data collection, preprocessing, expert annotation, and formatting for machine learning. The methodology ensures reproducibility, privacy compliance, and robustness across seasonal variations, addressing gaps in prior aerial datasets.

### 3.1 DATA COLLECTION AND PREPROCESSING

Data collection was performed using a DJI Phantom 4 RTK drone equipped with a 20-megapixel RGB camera capable of capturing images at a resolution of 5472×3648 pixels. The drone was flown at a consistent altitude of approximately 60 to 90 meters above ground level to optimize the trade-off between spatial coverage and image resolution (Ariram et al., 2024). Flights targeted key intersections types: 4-leg (crossroads) or 3-leg intersections (T/Y junctions) and roundabouts, during summer and winter to include diverse conditions such as snow, varying lighting, and weather. A total of around 1500 raw images were initially captured.

From the initial set of images, a manual filtering process was applied to remove irrelevant images (e.g., those without intersections) and low-quality samples (e.g., blurry or improperly exposed). However, 62 background samples (e.g., images without any classes of interest) were intentionally retained to improve model robustness against false positives, as suggested in prior work by Nogueira-Rodríguez et al. (2023). This resulted in a curated dataset of 969 relevant images containing clearly visible intersections. For training, images were resized to 960×640 (3:2 aspect ratio) for YOLO-based models, following best practices Redmon & Farhadi (2018). No color correction was applied to preserve raw aerial characteristics.

### 3.2 DATA ANNOTATION

Annotations were performed using the CVAT (Computer Vision Annotation Tool) application Sekachev et al. (2020), which was deployed on a local machine to ensure data privacy and maintain complete control over the annotation process. Two classes were defined: "roundabout" (class 0) and "intersection" (class 1, encompassing T/Y and 4-leg types). Bounding boxes were drawn around each instance of these classes in the images following the unified boundary definitions as mentioned in Appendix A. The annotation process was carried out, following a standardized protocol and reviewed by another annotator to ensure consistency and accuracy (Ahmadzadeh et al., 2025), following common quality-control practices used in large-scale datasets (Everingham et al., 2010; Lin et al., 2015). To ensure consistency, an annotation guideline was followed prior to labeling, and a two-pass review process was implemented as detailed in Appendix B.

The annotation protocol was designed according to a data minimization principle: in addition to the raw images required for intersection detection, we only store geometry level labels for road features (bounding boxes and class IDs for intersections or roundabouts) and collect no extra person or vehicle specific attributes or identifiers. The workflow also omits additional per-instance metadata such as weather or visibility flags, per-instance occlusion tags, or amodal bounding boxes, as detailed in Appendices C and D; aligning with GDPR recommendations to limit processing to what is necessary for the declared research purpose. Some of the annotated examples are shown in Figure 2.

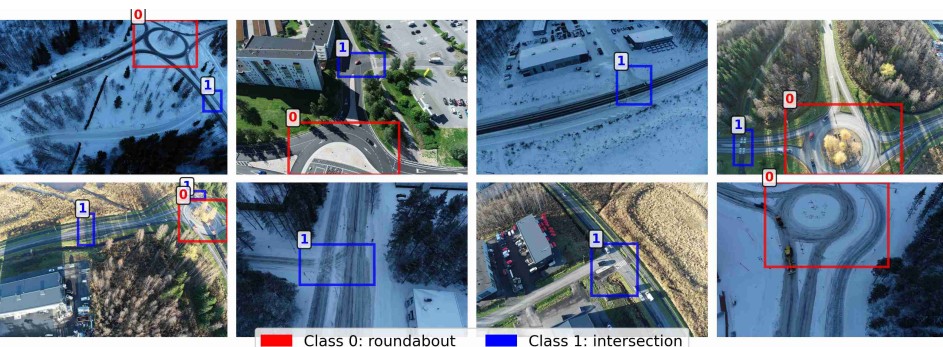

Figure 2: Example annotated images showing bounding boxes for roundabouts and intersections.

## 3.3 DATA SPLITTING

Proper dataset splitting is essential to prevent overfitting and ensure unbiased evaluation of model generalization Goodfellow et al. (2016). The training set is used for optimizing model parameters, the validation set facilitates hyperparameter tuning and early stopping, and the test set provides a final, unbiased estimate of model performance. A split ratio of 70:15:15 was chosen, following widely adopted conventions in computer vision and machine learning benchmarks, where 70-80% of data is allocated for training and the remainder is split between validation and testing Joseph (2022).

The split was performed using scikit-learn's `train_test_split` function, initially separating 70% of the data for training, followed by an equal division of the remaining 30% between validation and testing sets. This approach ensured strict independence between subsets, preventing data leakage. The use of stratified random sampling with a fixed random seed (42) guarantees reproducible results across experimental runs. As a result, the dataset was divided into subsets of 678, 145, and 146 images for training, validation, and testing respectively, which is detailed in Table 4.2. Additionally, a comprehensive algorithm procedure for the splitting process is provided in Appendix 1.

## 3.4 DATASET FORMATTING

The dataset was formatted following the requirements of the YOLO family of models, ensuring direct compatibility with modern training pipelines. Each image is stored in the `.jpg` format, and each corresponding label is saved as a `.txt` file with the same base filename. Labels follow the YOLO convention, where each line in the file corresponds to one annotated object using the structure: `class_id x_center y_center width height`. Here, all coordinates are normalized to the range $[0, 1]$ with respect to the image dimensions. For example, `0 0.5 0.5 0.25 0.25` would indicate an object of class ID 0 located at the center of the image with width and height equal to 25% of the image size Gope et al. (2024).

The dataset directory is divided into `train/`, `val/`, and `test/`, each containing two subfolders: `images/` (RGB `.jpg` files) and `labels/` (YOLO `.txt` annotations). The overall folder structure is shown in Appendix F. To enable training, a configuration file (`data.yaml`) defines dataset paths and class information. It specifies the locations of the `train`, `val`, and `test` sets, along with the number and names of classes. The complete file is provided in the Appendix G. This formatting ensures interoperability with YOLOv8, YOLOv11 as well as YOLOv12 versions. In practice, such

standardized formatting reduces preprocessing overhead, supports reproducibility, and aligns with best practices in dataset sharing Yang et al. (2023).

# 4 DATASET STATISTICS

This section presents comprehensive statistics for the ROADSIGHT dataset. Table 1 summarizes key attributes including image counts, annotations, classes, formats, and resolution. Subsequent sections detail class distribution, bounding box characteristics, and data splitting.

Table 1: Dataset overview including general attributes and image properties

| Attribute | Value |
|---|---|
| Total images | 969 (907 with classes, 62 background) |
| Total annotations | 1,355 |
| Classes | 2 (Roundabout, Intersection) |
| Image format | JPEG |
| Annotation format | YOLO (normalized coordinates) |
| Image resolution | 960 × 640 pixels |
| Aspect ratio | 3:2 (1.5:1) |
| Color space | RGB (Red Green Blue) |
| Image size (in KB) | Min: 68, Max: 599, Avg: 182 |

## 4.1 CLASS AND BOUNDING BOX STATISTICS

The dataset exhibits distinct bounding box characteristics across the two classes, as detailed in Table 2. Roundabouts generally occupy larger portions of the image, with mean width ($0.413 \pm 0.173$), height ($0.488 \pm 0.212$), and area ($0.223 \pm 0.162$) significantly higher than intersections (width: $0.210 \pm 0.158$, height: $0.210 \pm 0.114$, area: $0.055 \pm 0.078$). This reflects the typically larger physical footprint of roundabouts compared to intersections. Aspect ratios show moderate variability for both classes, with roundabouts averaging $0.959 \pm 0.549$ and intersections $1.104 \pm 0.673$, indicating a tendency toward more square-like shapes for roundabouts and slightly elongated forms for intersections. The ranges highlight the diversity within each class, with intersections displaying greater variability in width and aspect ratio, potentially due to the inclusion of various intersection geometries (3-leg and 4-leg). These statistics underscore the dataset's representation of real-world road features and inform model training by highlighting class-specific scale differences.

Table 2: Bounding box statistics for each class showing width, height, area, and aspect ratio distributions with normalized coordinates (0-1 range).

| Metric | Roundabout | | | Intersection | | |
|---|---|---|---|---|---|---|
| | Mean ± Std | Median | Range | Mean ± Std | Median | Range |
| Width | 0.413 ± 0.173 | 0.450 | 0.027 - 0.749 | 0.210 ± 0.158 | 0.163 | 0.013 - 0.908 |
| Height | 0.488 ± 0.212 | 0.437 | 0.092 - 0.950 | 0.210 ± 0.114 | 0.191 | 0.026 - 0.737 |
| Area | 0.223 ± 0.162 | 0.172 | 0.008 - 0.593 | 0.055 ± 0.078 | 0.029 | 0.002 - 0.501 |
| Ratio | 0.959 ± 0.549 | 0.792 | 0.095 - 3.315 | 1.104 ± 0.673 | 0.990 | 0.082 - 5.031 |

## 4.2 DATA SPLIT DISTRIBUTION

Table 3 provides a detailed breakdown of the distribution of both images and annotated instances across the training, validation, and testing sets, as described in Section 3.3. The results confirm that the 70:15:15 split ratio effectively preserves class proportions, with intersections accounting for 55.7-57.3% and roundabouts representing 42.7-44.3% consistently across all subsets.

Table 3: Dataset split distribution showing image counts and class-specific instance allocation across training, validation, and testing sets.

| Split | Images | Intersections | Roundabouts | Total Instances |
|---|---|---|---|---|
| Training | 678 (69.9%) | 540 (70.1%) | 402 (68.7%) | 942 (69.5%) |
| Validation | 145 (15.0%) | 112 (14.5%) | 89 (15.2%) | 201 (14.8%) |
| Testing | 146 (15.1%) | 118 (15.3%) | 94 (16.1%) | 212 (15.6%) |
| **Total** | **969 (100%)** | **770 (100%)** | **585 (100%)** | **1,355 (100%)** |

## 5 ALGORITHMIC ANALYSIS

This section presents a comprehensive evaluation of modern YOLO and DETR architectures on the ROADSIGHT dataset, focusing on establishing robust benchmarks for road intersection detection in UAV imagery. The analysis encompasses experimental setup configuration, model selection rationale, performance evaluation across multiple models, and edge device deployment optimization. The trade-offs between detection accuracy and computational efficiency are systematically evaluated, providing detailed performance metrics and deployment feasibility analysis.

### 5.1 EXPERIMENTAL SETUP

Experiments were conducted on both a dedicated high-end workstation and a resource-constrained edge device. Model training and initial evaluation were performed on a workstation equipped with an NVIDIA GeForce RTX 2080 Ti GPU (11GB GDDR6, 4352 CUDA cores), an Intel Core i7-8700 CPU (3.20 GHz, 6 cores/12 threads), and 64 GB RAM (DDR4). The software stack included Python 3.10.12 for scripting, PyTorch 2.8.0 for deep learning, CUDA 12.8 for GPU acceleration, Ultralytics YOLO v8.3.189 for model implementation, WandB 0.21.3 for experiment tracking, and Ubuntu 22.04 as the operating system.

For deployment, inference was conducted on an NVIDIA Jetson Orin Nano edge device equipped with an ARM Cortex-A78AE CPU, a 1024-core integrated GPU, 8 GB LPDDR5 memory, and delivering 40 TOPS of AI performance in a 15 W power mode.

### 5.2 MODEL SELECTION AND TRAINING PARAMETERS

The YOLO family was selected as the primary detection framework due to its native compatibility with the dataset format and proven effectiveness in real-time object detection tasks. To comprehensively evaluate the dataset's robustness and establish performance benchmarks, three popular YOLO variants were employed: YOLOv8, YOLOv11, and YOLOv12. Multiple model sizes, such as nano(n), small(s), medium(s), large(l), and extra-large(x) were evaluated to analyze the trade-offs between computational efficiency and detection accuracy. The nano variants prioritize speed and low resource consumption, making them suitable for edge deployment, while larger variants offer higher accuracy at the cost of increased computational requirements. Batch sizes of 16 were used across experiments, with some larger models limited to smaller batch sizes of 8 due to GPU memory constraints on the GPU memory. Additionally, some large models such as, YOLOv12l and YOLOv12x, were excluded from certain experiments due to excessive memory demands, which are not feasible for edge deployment scenarios.

Training was conducted for 100 epochs with an early stopping patience of 15 epochs to prevent overfitting. The optimizer was set to automatic selection, with an initial learning rate of 0.01, final learning rate of 0.01, momentum of 0.937, and weight decay of 0.0005. Automatic Mixed Precision (AMP) (Micikevicius et al., 2018) training was enabled to improve memory efficiency and training speed. Loss function 1 weighting was optimized for the dataset's class distribution, with classification loss weight of 0.5 and bounding box regression loss weight of 7.5. Data augmentation techniques included geometric transformations (rotation, translation, scaling, shearing), photometric adjustments (hue, saturation, brightness variations) and advanced augmentation strategies such as mosaic, mixup, copy-paste, and random erasing. These techniques were specifically tuned for aerial road intersection imagery to improve model robustness across different seasonal and light-

ing conditions. Detailed hyperparameter values for all augmentation techniques are provided in Appendix I Table 7). These values follow the recommended Ultralytics defaults, which have been empirically validated across a wide range of YOLO architectures and datasets, providing a stable and well-balanced optimization configuration for object detection tasks (Lee et al., 2025; Yu & Zhou, 2023).

Model evaluation followed a rigorous multi-run validation protocol to ensure statistical reliability. Performance assessment utilized 11 independent runs with the first run excluded for JIT (Just-In-Time) compilation warm-up (Ansel et al., 2024), to achieve accurate GPU-accelerated inference benchmarking. All reported performance metrics represent averages computed from the remaining 10 runs to ensure robust and reliable measurements. Standard object detection metrics were employed, including mAP 8 at IoU 2 thresholds of 0.5 (mAP50 9) and 0.5 to 0.95 (mAP50-95 10), along with precision 3, recall 4, and F1-score 5. Comprehensive definitions and mathematical formulations for all performance metrics are provided in Appendix H (Sapkota et al., 2025; He et al., 2024). The validation set was used for hyperparameter tuning and model selection, while the test set provided final performance evaluation to ensure unbiased assessment of the dataset's effectiveness across different YOLO architectures.

In addition to the YOLO family, we included real-time Detection Transformer (RT-DETR) variants (RT-DETR-l and RT-DETR-x) to further validate the dataset under a complementary architectural paradigm. DETR style models rely on transformer-based encoder-decoder mechanisms and bipartite matching loss (Carion et al., 2020), providing a contrast to anchor-based, one-stage detectors such as YOLO. To ensure a fair comparison focused on dataset characteristics rather than optimizer tuning, RT-DETR models were trained with the same core optimization settings as the YOLO experiments (100 epochs, early-stopping patience of 15, initial learning rate 0.01, momentum 0.937, weight decay 0.0005, and AMP enabled), while using the Ultralytics-provided RT-DETR training configuration for loss components and scheduler details. Data augmentation followed the same policy described above, so that performance differences primarily reflect architectural behaviour on ROADSIGHT rather than differing training recipes.

## 5.3 PERFORMANCE EVALUATION METRICS

Table 4 presents a comprehensive comparison of model performance across YOLOv8, YOLOv11, YOLOv12, and DETR variants with different variants and batch configurations. The evaluation encompasses key object detection metrics, including mAP50, mAP50-95, precision, recall, F1-score, and inference speed measured in milliseconds per image.

YOLOv11s with batch size 16 emerged as the optimal model, achieving the highest mAP50-95 of 0.676 while maintaining efficient inference speed of 6.668ms. While nano variants (YOLOv8n, YOLOv11n) delivered faster inference (4.653-6.870ms), they achieved lower mAP50-95 values (0.635-0.637), and larger variants (YOLOv8x, YOLOv11x) suffered from significantly slower inference times (18.371-18.507ms) despite comparable accuracy. Notably, YOLOv11n achieved superior mAP50 performance (0.975) compared to YOLOv11s (0.966), but struggled with the more stringent mAP50-95 (0.637 vs 0.676), highlighting the importance of consistent performance across IoU thresholds for robust detection. In comparison, DETR variants (RT-DETR-l and RT-DETR-x) showed lower mAP50-95 scores (0.599 and 0.567, respectively) with moderate inference times (10.6-13.9ms), indicating that while they provide competitive baselines, they do not surpass the YOLOv11s in overall accuracy-efficiency trade-off. YOLOv11s strikes the optimal balance between detection accuracy and computational efficiency, making it ideal for real-time UAV intersection detection applications.

## 5.4 ANALYSIS OF BEST PERFORMING MODEL

Figure 3 presents a comprehensive analysis of the YOLOv11s model performance through multiple visualization components. The F1-confidence curve (A) shows optimal overall performance with all classes achieving F1-score of 0.95 at confidence threshold 0.392, indicating robust detection capabilities across both roundabout and intersection classes. The precision-recall curve (B) demonstrates excellent class-specific performance with roundabouts achieving 0.995 precision and intersections reaching 0.937 precision, while maintaining high recall rates. The precision-confidence curve (C) reveals stable performance across varying confidence thresholds, with both classes maintaining pre-

Table 4: Model performance comparison across YOLOv8, YOLOv11, YOLOv12, and DETR variants with different variants and batch sizes.

| Model | Batch | mAP50 | mAP50-95 | Precision | Recall | F1 | Inference(ms) |
|---|---|---|---|---|---|---|---|
| YOLOv8n | 16 | 0.969 | 0.635 | 0.978 | 0.956 | 0.967 | 4.653 |
| YOLOv8s | 16 | 0.974 | 0.661 | 0.995 | 0.941 | 0.968 | 4.885 |
| YOLOv8m | 16 | 0.963 | 0.657 | 0.984 | 0.921 | 0.951 | 7.366 |
| YOLOv8l | 16 | 0.965 | 0.629 | 0.978 | 0.939 | 0.958 | 12.707 |
| YOLOv8x | 8 | 0.966 | 0.636 | 0.992 | 0.926 | 0.958 | 18.371 |
| YOLOv11n | 16 | 0.975 | 0.637 | 0.979 | 0.940 | 0.959 | 6.870 |
| YOLOv11s | 16 | 0.966 | 0.676 | 0.958 | 0.937 | 0.947 | 6.668 |
| YOLOv11m | 16 | 0.949 | 0.610 | 0.970 | 0.882 | 0.924 | 8.707 |
| YOLOv11l | 16 | 0.968 | 0.640 | 0.982 | 0.922 | 0.951 | 12.192 |
| YOLOv11x | 8 | 0.957 | 0.608 | 0.993 | 0.901 | 0.945 | 18.507 |
| YOLOv12n | 16 | 0.970 | 0.648 | 0.980 | 0.932 | 0.955 | 10.304 |
| YOLOv12s | 16 | 0.961 | 0.649 | 0.967 | 0.949 | 0.958 | 10.461 |
| YOLOv12m | 8 | 0.968 | 0.614 | 0.953 | 0.947 | 0.950 | 11.169 |
| RT-DETR-l | 8 | 0.966 | 0.599 | 0.948 | 0.964 | 0.956 | 10.6 |
| RT-DETR-x | 8 | 0.919 | 0.567 | 0.904 | 0.860 | 0.882 | 13.9 |

cision above 0.9 for confidence values up to 0.8. The recall-confidence curve (D) shows consistent recall performance, with all classes achieving recall above 0.97 at low confidence thresholds.

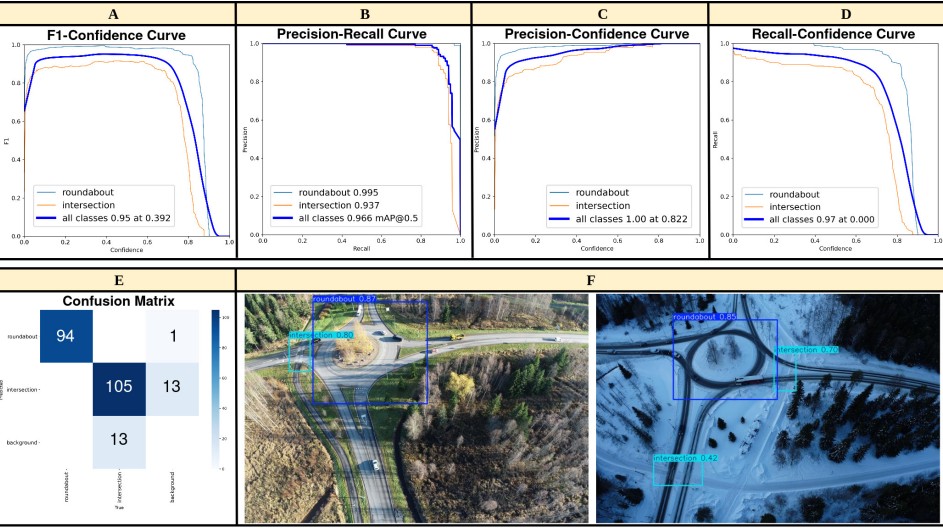

Figure 3: Comprehensive performance analysis of the best performing YOLOv11s model with batch size 16. (A) F1-confidence curve, (B) Precision-recall curve, (C) Precision-confidence curve, (D) Recall-confidence curve, (E) Confusion matrix, and (F) Qualitative detection examples under different seasonal conditions.

The confusion matrix (E) provides detailed insight into classification performance, showing minimal misclassification between roundabouts and intersections. The matrix reveals that the model correctly identifies the vast majority of instances for both classes, with very few false positives or false negatives, demonstrating the dataset's effectiveness in training robust intersection detection models. Qualitative analysis through detection examples (F) showcases the model's capability to accurately detect intersections under diverse environmental conditions. The left image demonstrates successful roundabout and intersection detection in summer conditions with a high confidence of 0.87 and 0.8 respectively, while the right image shows robust performance in winter conditions with snow coverage, successfully detecting both roundabouts (0.56) and intersections (0.70, 0.42) de-

spite challenging visibility conditions. These results validate the dataset's seasonal diversity and the model's generalization capability across different weather conditions, lighting variations, and intersection geometries.

## 5.5 K-Fold Cross-Validation Analysis

To further validate the robustness and generalizability of the YOLOv11s model on the ROADSIGHT dataset, we conducted a 5-fold cross-validation using the same hyperparameters as Section 5.2. The results demonstrate consistent performance across folds, with average metrics of 0.950 precision, 0.925 recall, 0.937 F1-score, 0.962 mAP50, and 0.669 mAP50-95. Full per-fold results, including detailed metrics for each of the five folds are given in Table 5. This indicates low variance in model performance, confirming the reliability of YOLOv11s for intersection detection under varying data subsets.

Table 5: K-Fold cross-validation performance of YOLOv11s on the proposed dataset (5 Folds).

| Fold | Images | Instances | Precision | Recall | F1 | mAP50 | mAP50-95 |
|------|--------|-----------|-----------|--------|-------|-------|----------|
| 1 | 182 | 281 | 0.945 | 0.917 | 0.931 | 0.963 | 0.653 |
| 2 | 182 | 253 | 0.950 | 0.935 | 0.942 | 0.971 | 0.676 |
| 3 | 181 | 259 | 0.932 | 0.922 | 0.927 | 0.955 | 0.677 |
| 4 | 181 | 268 | 0.961 | 0.921 | 0.941 | 0.960 | 0.670 |
| 5 | 181 | 294 | 0.963 | 0.928 | 0.945 | 0.963 | 0.669 |
| **Avg** | **181.4** | **271.0** | **0.950** | **0.925** | **0.937** | **0.962** | **0.669** |

## 5.6 Edge Device Benchmarking

To evaluate the practical deployment feasibility of the trained YOLOv11s model on resource-constrained edge devices, comprehensive benchmarking was conducted on the NVIDIA Jetson Orin Nano platform. The evaluation focused on analyzing the trade-offs between inference speed, detection accuracy, and memory consumption across different model formats such as Pytorch (Paszke et al., 2019), TorchScript (DeVito, 2022), ONNX (Bai et al., 2019), TensorRT (Jeong et al., 2022), Tensorflow (Abadi et al., 2015), MNN (Jiang et al., 2020), NCNN (Ni & The ncnn contributors, 2017) and quantization schemes (FP32, FP16 and INT8) (Jacob et al., 2017). Table 6 presents the performance comparison of various export formats and quantization levels, providing insights into optimal performance for real-time UAV-based intersection detection.

The benchmarking results demonstrate that TensorRT with FP16 quantization provides the optimal configuration for real-time UAV deployment, achieving 14.5ms inference speed (69 FPS) while maintaining high accuracy with mAP50-95 of 0.664. Although the INT8 variant offers faster inference at 10.9ms (92 FPS), it compromises accuracy with mAP95 dropping to 0.653, while FP16 delivers only minimal accuracy degradation compared to the PyTorch FP32 baseline (0.684) with a significant 55% speed improvement from 32.6ms. The rest of the frameworks, such as Tensorflow, ONNX, MNN and NCNN exhibit substantially slower performance, making them unsuitable for real-time applications. The TensorRT FP16 configuration emerges as the preferred choice for real-time intersection detection because it strikes the critical balance between maintaining detection accuracy required for reliable UAV navigation while achieving inference speeds exceeding 60 FPS necessary for real-time processing, making it ideal for resource-constrained edge deployment scenarios.

## 6 Limitations

While ROADSIGHT advances UAV-based intersection detection, several limitations remain. The dataset's scale is modest, collected solely from localized Finnish areas, potentially limiting generalization to diverse global road infrastructures and climates. Environmental diversity is constrained, primarily covering summer and winter with limited representation of adverse conditions like heavy rain, fog, or active snowfall, which could affect robustness in extreme scenarios. The classification

Table 6: Edge device performance comparison across different model formats and quantization schemes of trained YOLOv11s on NVIDIA Jetson Orin Nano

| Format | Size (MB) | F1 | mAP50 | mAP50-95 | Inference (ms) |
|---|---|---|---|---|---|
| PyTorch (FP32) | 18.3 | 0.963 | 0.971 | 0.684 | 32.6 |
| PyTorch (FP16) | 18.3 | 0.963 | 0.971 | 0.686 | 31.9 |
| PyTorch (INT8) | 18.3 | 0.963 | 0.971 | 0.684 | 33.9 |
| TorchScript (FP32) | 36.4 | 0.957 | 0.973 | 0.665 | 37.8 |
| TorchScript (FP16) | 36.4 | 0.957 | 0.973 | 0.664 | 26.9 |
| ONNX (FP16) | 18.1 | 0.957 | 0.974 | 0.666 | 342.5 |
| TensorRT (FP32) | 38.2 | 0.957 | 0.973 | 0.665 | 24.1 |
| TensorRT (FP16) | 21.7 | 0.957 | 0.973 | 0.664 | 14.5 |
| TensorRT (INT8) | 12.2 | 0.956 | 0.975 | 0.653 | 10.9 |
| TensorFlow (FP32) | 36.2 | 0.957 | 0.973 | 0.665 | 368.8 |
| MNN (INT8) | 9.3 | 0.961 | 0.974 | 0.663 | 212.1 |
| NCNN (FP16) | 18.2 | 0.958 | 0.973 | 0.666 | 326.5 |

granularity is simplified to two categories (roundabouts and intersections), merging 3-leg and 4-leg types, which may restrict applications requiring finer distinctions. Future work should expand data collection to encompass larger, more diverse geographic regions and environmental conditions and incorporate broader model comparisons to enhance dataset utility and robustness.

## 7 CONCLUSION

The ROADSIGHT dataset fills a critical gap in aerial computer vision benchmarks by providing unique, intersection-specific annotations across diverse seasonal variations, enabling robust model development for complex road geometries. Our comprehensive evaluation establishes baseline metrics for this task, demonstrating that modern architectures like YOLOv11s can achieve an optimal balance (0.676 mAP50-95, 6.668ms inference time) suitable for resource-constrained platforms. The successful deployment on NVIDIA Jetson Orin Nano at 69 FPS with TensorRT optimization further proves the practical viability of real-time intersection detection on edge hardware. By offering a specialized resource for seasonal adaptability and efficient inference, ROADSIGHT opens new possibilities for dynamic traffic monitoring, emergency response planning, and autonomous vehicle coordination, laying the groundwork for next-generation smart city infrastructure powered by intelligent UAVs.

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

## A   APPENDIX: UNIFIED BOUNDARY DEFINITIONS

To enable precise annotation, consistent modeling, and reliable evaluation of intersections and roundabouts in UAV-captured aerial imagery, it is necessary to establish unified boundary definitions (Sitran et al., 2016).

Intersection Boundary: We define an *intersection boundary* as the physical and functional region where two or more roads converge. This includes: (i) the *conflict area*, where vehicular paths cross, merge, or diverge; (ii) the *approach legs*, referring to the roadway segments leading into the junction; and (iii) *auxiliary elements* such as turning lanes, channelization features (e.g., splitter islands), and pedestrian facilities.

Roundabout Boundary: The *roundabout boundary* encompasses: (i) the inscribed circle diameter, representing the outer perimeter of the circulatory roadway; (ii) the central island; (iii) approach splitter islands; and (iv) the truck apron, when present.

## B   APPENDIX: ANNOTATION PROCESS

The annotation process followed a structured protocol to ensure high-quality, consistent labels for road intersections and roundabouts. The process consisted of the following steps:

1. **Guideline Development**: Prior to labeling, a detailed annotation guideline was established to standardize box placement and class assignment. The guideline specified: - Drawing bounding boxes as tightly as possible around the functional road geometry. - Including the entire navigable circular structure for roundabouts, even when vegetation or shadows were present. - Ensuring that intersection boxes covered all approaching legs that visually define the junction.

2. **Primary Annotation**: Annotations were first created by a primary annotator using CVAT, adhering to the guideline and unified boundary definitions (Appendix A).

3. **Secondary Review**: Each annotated image was subsequently reviewed by a second annotator. During this review: - Bounding boxes with noticeable deviations from the guideline, such as overly loose boxes or partially enclosed roundabouts, were corrected. - Cases with disagreement on class assignment were discussed and resolved to reach consensus.

4. **Quality Assurance**: The two-pass process ensured the final dataset adhered to standardized annotation rules, minimizing inconsistencies and improving overall accuracy.

## C   APPENDIX: PRIVACY AND GDPR ANONYMIZATION NOTE

This release does not include per-object annotations of vehicles/pedestrians. Future versions will: - add anonymization of identifiable features (faces, license plates) via blurring/masking before annotation, - maintain an audit trail of anonymization steps, - store only geometry-focused labels for intersections. Processing complies with EU GDPR principles (data minimization, purpose limitation, privacy by design).

## D   APPENDIX: OCCLUSION AND METADATA PLAN

To address dynamic occlusion in future releases, we plan to add: - per-image weather/visibility flags (sunny, overcast, light snow cover, heavy snow, rain, fog), - per-instance occlusion flags (none/partial/heavy), - optional amodal bounding boxes to capture full extent when partially occluded. These metadata will be documented in the dataset card and YAML schema for downstream training.

## E   APPENDIX: DATA SPLITTING IMPLEMENTATION

The dataset splitting and organization procedure is outlined in Algorithm 1. This algorithm ensures a reproducible and systematic approach to dividing the dataset into training, validation, and test sets, while maintaining the integrity of image-label pairs as described in Section 3.3.

---

**Algorithm 1:** Dataset Splitting and Organization Procedure

---

**Input:** Directory containing images and labels
**Output:** Train, validation, and test sets with corresponding labels
1 Get all image filenames ending with '.jpg' from dataset directory;
2 Split images into 70% training and 30% temporary using fixed random seed;
3 Split the temporary 30% equally into 15% validation and 15% test sets;
4 **foreach** *split in [train, val, test]* **do**
5   |   Create 'images/' and 'labels/' subdirectories if not existing;
6 **end**
7 **foreach** *file in each split set* **do**
8   |   Move image to corresponding 'images/' subfolder;
9   |   If label file exists (.txt), move it to 'labels/' subfolder;
10 **end**

---

# F  APPENDIX: DATA FOLDER SCHEMA

```
path_to_data/
    |-- train/
    |    |-- images/ (678 samples)
    |    |-- labels/ (634 samples)
    |-- val/
    |    |-- images/ (145 samples)
    |    |-- labels/ (137 samples)
    |-- test/
         |-- images/ (146 samples)
         |-- labels/ (136 samples)
```

# G  APPENDIX: DATASET YAML CONFIGURATION

```
train: ./train
val: ./val
test: ./test
nc: 2
names: ["roundabout", "intersection"]

# Class weights calculated from inverse frequency
# roundabout: 1355/585 = 2.32, intersection: 1355/770 = 1.76
# Normalized: roundabout: 1.32, intersection: 1.0
cls: [1.32, 1.0]
```

# H  APPENDIX: PERFORMANCE METRICS DEFINITION

This appendix provides comprehensive mathematical definitions for all performance metrics used in the evaluation protocol discussed in Section 5.2. The metrics follow standard object detection evaluation protocols and are essential for understanding the quantitative assessment of model performance.

**YOLOv11 Loss Function**: The YOLOv11 loss function combines three components for optimized object detection performance:

$$L = L_{cls} + L_{box} + L_{dfl} \tag{1}$$

where $L_{cls}$ is the class probability loss based on cross-entropy for classification accuracy, $L_{box}$ is the bounding box regression loss using IoU metrics for localization precision, and $L_{dfl}$ is the distributed focal loss that prioritizes challenging samples to improve overall detection performance.

**Intersection over Union (IoU)**: The Intersection over Union (IoU) is a fundamental metric for measuring the overlap between predicted and ground truth bounding boxes. It is defined as:

$$\text{IoU} = \frac{\text{Area of Intersection}}{\text{Area of Union}} = \frac{|B_{\text{pred}} \cap B_{\text{gt}}|}{|B_{\text{pred}} \cup B_{\text{gt}}|} \tag{2}$$

where $B_{\text{pred}}$ represents the predicted bounding box and $B_{\text{gt}}$ represents the ground truth bounding box. IoU values range from 0 (no overlap) to 1 (perfect overlap).

**Precision and Recall**: Precision measures the fraction of positive predictions that are actually correct, while recall measures the fraction of actual positives that are correctly identified.

$$\text{Precision} = \frac{\text{TP}}{\text{TP} + \text{FP}} \tag{3}$$

$$\text{Recall} = \frac{\text{TP}}{\text{TP} + \text{FN}} \tag{4}$$

where TP (True Positives) are correctly detected objects with IoU $\geq$ threshold, FP (False Positives) are incorrectly detected objects with IoU $<$ threshold, and FN (False Negatives) are ground truth objects that were not detected.

**F1-Score**: The F1-Score is the harmonic mean of precision and recall, providing a single metric that balances both measures:

$$\text{F1-Score} = 2 \times \frac{\text{Precision} \times \text{Recall}}{\text{Precision} + \text{Recall}} \tag{5}$$

**Average Precision (AP)**: Average Precision summarizes the precision-recall curve as the area under the curve. For a single class, AP is computed as:

$$\text{AP} = \int_0^1 P(R) \, dR \tag{6}$$

where $P(R)$ is the precision as a function of recall $R$. In practice, this is approximated using interpolated precision values:

$$\text{AP} = \sum_{k=1}^{N} (R_k - R_{k-1}) \times P_{\text{interp}}(R_k) \tag{7}$$

where $P_{\text{interp}}(R_k) = \max_{R' \geq R_k} P(R')$ ensures a monotonically decreasing precision curve.

**Mean Average Precision (mAP)**: Mean Average Precision is the average of AP values across all classes:

$$\text{mAP} = \frac{1}{C} \sum_{c=1}^{C} \text{AP}_c \tag{8}$$

where $C$ is the number of classes and $\text{AP}_c$ is the average precision for class $c$.

**mAP50** (also known as mAP@0.5) uses a single IoU threshold of 0.5 to determine true positive detections:

$$\text{mAP50} = \frac{1}{C} \sum_{c=1}^{C} \text{AP}_c^{0.5} \tag{9}$$

**mAP50-95** (also known as mAP@0.5:0.95) averages mAP values across multiple IoU thresholds from 0.5 to 0.95 with a step size of 0.05:

$$\text{mAP50-95} = \frac{1}{10}\sum_{t=1}^{10}\text{mAP@}(0.45 + 0.05t) \tag{10}$$

This metric provides a more comprehensive evaluation as it requires higher localization accuracy for detections to be considered correct.

## I    APPENDIX: AUGMENTATION HYPERPARAMETERS DURING TRAINING

Table 7: Augmentation hyperparameters used in YOLOv8, YOLOv11 and YOLOv12 training.

| Augmentation Hyperparameters (`argument`) | Value |
|---|---|
| Type of auto-augmentation policy (`auto_augment`) | randaugment |
| Range of rotation for augmentation (`degrees`) | 10.00 |
| Hue augmentation factor (`hsv_h`) | 0.015 |
| Saturation augmentation factor (`hsv_s`) | 0.700 |
| Brightness augmentation factor (`hsv_v`) | 0.400 |
| Translation factor for image shift (`translate`) | 0.100 |
| Scaling factor for image resizing (`scale`) | 0.200 |
| Shear transformation range (`shear`) | 5.00 |
| Horizontal flip probability (`fliplr`) | 0.500 |
| Vertical flip probability (`flipud`) | 0.000 |
| Mixup augmentation probability (`mixup`) | 0.150 |
| Mosaic augmentation probability (`mosaic`) | 0.700 |
| Probability of random erasing (`erasing`) | 0.400 |
| Copy-paste augmentation probability (`copy_paste`) | 0.300 |
| Copy-paste mode (`copy_paste_mode`) | flip |
| Perspective distortion level (`perspective`) | 0.000 |
| Probability of BGR channel shuffling (`bgr`) | 0.000 |
| CutMix augmentation probability (`cutmix`) | 0.000 |

## J    APPENDIX: LARGE LANGUAGE MODEL USAGE STATEMENT

In accordance with ICLR 2026 submission guidelines, we disclose the use of Large Language Models (LLMs) in the preparation of this manuscript. Specifically, we used LLMs for the following purposes:

**Text Polishing and Enhancement**: LLMs were employed to improve the clarity, coherence, and academic writing style of various sections throughout the paper. This included grammar correction, sentence restructuring for better readability, and refinement of technical explanations while preserving the original scientific content and contributions.

**Code Formatting and Documentation**: LLMs assisted in formatting code snippets, algorithm descriptions, and improving the presentation of technical implementation details. This included standardizing code formatting conventions, enhancing code comments, and ensuring consistency in algorithmic pseudo-code presentation.

All scientific content, experimental design, data collection, analysis, and conclusions represent the original work and intellectual contributions of the authors. The use of LLMs was limited to improving the presentation and formatting quality of the manuscript without altering the fundamental scientific contributions or claims.

