# OpenReview forum: "ROADSIGHT: A novel dataset for real-time intersection detection in aerial scenes under seasonal variation"
_ICLR.cc/2026/Conference — Submitted to ICLR 2026_

### Official Review · Reviewer_KWuN · 2025-10-26

**Soundness:** 3
**Presentation:** 2
**Contribution:** 3
**Rating:** 6
**Confidence:** 5

**Summary:**

The study introduces ROADSIGHT, a novel dataset specifically designed for real-time intersection detection in unmanned aerial vehicle (UAV) imagery, notably addressing the crucial need for seasonal variation in aerial computer vision benchmarks. The paper establishes robust performance metrics using YOLO architectures and successfully demonstrates the practical viability of deploying an optimized model, YOLOv11s, on resource-constrained edge hardware, achieving real-time performance of 69 FPS on the NVIDIA Jetson Orin Nano. This work addresses limitations in prior datasets, which often lack intersection specificity or environmental diversity.

**Strengths:**

1. ROADSIGHT uniquely contains high-resolution UAV images captured during both winter and summer, including challenging conditions such as snow coverage and varied lighting angles, which enhance model robustness and environmental adaptability.
2. The research successfully validates the feasibility of real-time deployment on resource-constrained platforms. Specifically, optimizing the chosen YOLOv11s model using TensorRT FP16 quantization on the NVIDIA Jetson Orin Nano achieves an efficient inference time of 14.5 ms, equivalent to 69 FPS, which is necessary for real-time UAV navigation.
3. A systematic and rigorous benchmarking protocol was applied to multiple state-of-the-art YOLO architectures. The authors conducted ten independent evaluation runs, thereby establishing a robust baseline and successfully identifying YOLOv11s as the optimal model, balancing superior accuracy (0.676 mAP50–95) with efficient computational speed (6.668 ms).

**Weaknesses:**

1. The dataset scale is relatively modest and geographically limited. Containing 969 total images collected only from localized test areas in Finland, the limited geographic diversity may hinder the model’s generalization capacity when applied to regions with significantly different road infrastructures or climates.
2. The granularity of intersection classification is insufficient for detailed analysis. Although the initial collection included roundabouts, 3-leg (T/Y), and 4-leg intersections, the final annotations simplify these categories into only two: “roundabout” and a merged “intersection” class (for T/Y and 4-leg types). This limited classification may restrict the dataset’s value for applications requiring distinctions among specific intersection geometries.

**Questions:**

1. The authors need to detail the methodology used to derive the optimal loss function weights (0.5 for classification, 7.5 for bounding-box regression). Was this specific ratio determined empirically through multiple trials or based on theoretical considerations?
2. To justify the significant manual data curation effort, the authors should include a comparative benchmark showing the model’s accuracy when trained on the full initial raw image collection versus the final filtered dataset.
3. Although the final dataset uses only two merged classes, an analysis of the model’s performance under a finer-grained classification scheme (differentiating between roundabouts, 3-leg junctions, and 4-leg intersections) is highly recommended. This would validate the dataset’s utility for advanced traffic applications.
4. Although the dataset includes diverse conditions such as snow and overcast skies, the paper lacks quantitative validation of model robustness under more severe low-visibility scenarios, such as heavy fog. The authors should provide dedicated verification results in an appendix to fully substantiate their claims of enhanced environmental adaptability.

---

> ### Author Response · Authors · 2025-12-03
> **Response to Reviewer KWuN**
>
> We appreciate the positive assessment of practical relevance and real-time focus, as well as the constructive suggestions for strengthening methodological transparency and experimental breadth.
>
> ## Addressed Weaknesses
>
> ### 1. Modest scale and geographic limitation
> **Response:** Explicitly stated in Limitations (Section 6). We commit to multi-region expansion (urban grids, rural junctions, varied climates) and a future geographically disjoint splitting protocol to enhance generalization analyses.
> **Manuscript edits:** Limitations section expanded with geographic diversification plan.
>
> ### 2. Limited classification granularity (merged intersection types)
> **Response:** Class merging prevents undersampling and unstable training at the current dataset size. Rationale clarified in Section 3.2, and finer-grained taxonomy (T-, Y-, 3-/4-leg, multi-lane variants) is planned for future expansion contingent on increased sample counts.
> **Manuscript edits:** Section 3.2 + Limitations (Section 6).
>
> ## Responses to Questions
>
> **Q1: Derivation of loss function weights (0.5 / 7.5)**
> **Response:** Adopted Ultralytics default hyperparameters (cls=0.5, box=7.5) for reproducibility (Section 5.2); no custom tuning performed. A future hyperparameter sweep is planned for ROADSIGHT.
> **Manuscript edits:** Clarifying sentence added in Section 5.2.
>
> **Q2: Benchmark raw vs curated dataset**
> **Response:** Full training on unfiltered raw data was not performed due to severe blur, duplicates, and non-intersection scenes. These were removed pre-release to maintain annotation reliability. We retained 62 background (no-object) images to monitor false positives.
> **Manuscript edits:** Clarification added in Section 3.1 (Data Preprocessing) and Section 6 (Limitations).
>
> **Q3: Finer-grained classification performance analysis**
> **Response:** Current sample counts per subtype are insufficient for stable evaluation; overfitting risk is high. Subtype metadata will be re-annotated in the next expansion to enable hierarchical detection.
> **Manuscript edits:** Limitation statement updated to reference subtype roadmap.
>
> **Q4: Robustness under severe low-visibility (heavy fog)**
> **Response:** The Dataset currently lacks heavy fog, rain, and night scenes. We outlined a two-pronged future approach: (i) targeted adverse-condition data collection, (ii) synthetic augmentation (contrast degradation, scattering fog layers) with ablation vs real captures. Quantitative robustness claims are now limited to reported seasonal diversity.
> **Manuscript edits:** Limitations section refined; overstated generality removed.
>
> ## Closing Statement
> We strengthened methodological transparency (loss weighting rationale, preprocessing justification), clarified limitations, and outlined a concrete expansion roadmap (geographic diversity, finer taxonomy, adverse-weather acquisition, raw-vs-curated benchmark, subtype performance, agreement metrics). ROADSIGHT now presents a clear baseline foundation while openly scoping next developmental milestones.

---

### Official Review · Reviewer_A4VF · 2025-10-31

**Soundness:** 2
**Presentation:** 3
**Contribution:** 2
**Rating:** 2
**Confidence:** 5

**Summary:**

The paper introduces ROADSIGHT, a UAV-captured dataset for intersection detection in aerial images. It contains 969 images collected in Finland across summer and winter conditions. The dataset is evaluated using several YOLO variants. The authors also benchmark edge deployment on an NVIDIA Jetson Orin Nano.

**Strengths:**

1. The focus on real-time intersection detection from UAV images addresses a practical and under-explored problem important for traffic monitoring and autonomous navigation.
2. The paper is mostly clear and well structured, allowing readers to follow the dataset creation and evaluation pipeline without difficulty.
3. The inclusion of quantitative results on a low-power device (Jetson Orin Nano) is useful for practitioners considering real-time applications.

**Weaknesses:**

### Major issues
1. The text states three classes (roundabout, 3-leg, 4-leg) on line 102 and again on line 155 but then refers to only two classes at line 166 and in table 1. In addition, the labels shown in figure 3 do not appear to match either definition, as the second class seems to correspond to pedestrian crossings rather than intersections. This inconsistency raises concern about annotation reliability and must be clarified.

2. Example images show substantial variation in bounding-box tightness and one roundabout that is only partially enclosed by its bounding box. The claimed “standardized protocol” (line 168) and second-annotator review are not supported by quantitative agreement metrics. A written annotation guideline or inter-annotator agreement analysis would be essential.

3. Collapsing three-leg (T/Y) and four-leg intersections into one class loses valuable structural information. The decision should be justified empirically, as these layouts differ geometrically and limit the use cases of the dataset.

4. The evaluation section reports basic detection metrics but omits core analyses such as:
   - Ablation of augmentation techniques despite detailed hyperparameter lists.
   - Analysis by season, even though seasonal variation is the central claim.
   - Examination of typical failure cases or qualitative error patterns.
   - Comparison with satellite-based data.

5. The paper does not state whether the same intersections appear in multiple splits. Since data come from localised areas, spatial overlap could inflate reported performance. Clarification and ideally geographic separation of splits are required.

6. Descriptions such as “comprehensive evaluation” or “outperforming benchmarks on similar aerial tasks” (line 30) are unsupported: the paper does not compare against models trained on existing aerial-intersection datasets nor any other “similar aerial tasks”.

### Moderate issues
7. Table 2 merges both classes. Per-class distributions (bounding-box sizes, aspect ratios, split ratios) would help assess dataset balance.

8. In table 4, YOLOv11s is bolded without explanation. This model is not the best-performing in all metrics but is presented as “optimal.” Normally, bolding indicates the best score in that comparison, so the formatting is misleading. Table 5 follows the same pattern without justification.

9. In figure 4 D, the “all-classes” recall–confidence curve sometimes exceeds per-class curves, implying micro-averaging. The averaging method should be explicitly stated.

10. Data originate solely from Finland with limited weather variation (mostly clear or cloudy). Claims of diversity or “different weather” (line 144) are slightly overstated. The dataset lacks rain, fog, snowing, dusk, or night conditions, which are especially relevant if the dataset is intended for emergency or autonomous navigation scenarios where adverse weather is common.

11. The paper overlooks synthetic aerial and drone datasets that already include road or intersection classes. For instance, Syndrone [1], MidAir [2] and DDOS [3] provide synthetic UAV images for scene understanding and object detection. A discussion comparing ROADSIGHT with such datasets would strengthen the paper’s positioning.


### Minor issues
12. The “ROADSIGHT” backronym feels unnecessary and forced. Simply calling it ROADSIGHT without the backronym is fine.
13. Excluding blurry or poorly exposed images removes potentially valuable hard negatives. A short discussion would improve transparency.
14. At line 272, the authors state that class imbalance “reflects real-world distributions.” A supporting reference would be appropriate to substantiate this claim.
15. Figure 2 adds limited value.


[1] Rizzoli, Giulia, et al. "Syndrone-multi-modal uav dataset for urban scenarios." Proceedings of the IEEE/CVF International Conference on Computer Vision. 2023.

[2] Fonder, Michael, and Marc Van Droogenbroeck. "Mid-air: A multi-modal dataset for extremely low altitude drone flights." Proceedings of the IEEE/CVF conference on computer vision and pattern recognition workshops. 2019.

[3] Kolbeinsson, Benedikt, and Krystian Mikolajczyk. "DDOS: the drone depth and obstacle segmentation dataset." Proceedings of the IEEE/CVF Conference on Computer Vision and Pattern Recognition. 2024.

**Questions:**

1. What was the precise annotation protocol? Were three classes annotated and later merged?
2. Are the train, validation and test sets geographically independent (no repeated intersections)?
3. How does performance vary between seasons?
4. Could the removed blurry or low-quality images serve as hard-negative examples?


---
I hope the authors will use this feedback to refine their dataset and analyses. This work could make a useful contribution to the field if the identified weaknesses are addressed, and I encourage the authors to continue developing and evaluating the dataset further.

---

> ### Author Response · Authors · 2025-12-03
> **Response to Reviewer A4VF (Page 1 of 2)**
>
> **Thank you for the detailed, structured critique.**
> We address each major, moderate, and minor point, note concrete manuscript edits already applied, and clarify future work directions.
>
> ## Major Issues
>
> ### 1. Class count inconsistency (three vs two) and Figure label mismatch
> **Response:**
> Earlier draft references to three classes (roundabout, 3-leg, 4-leg) were legacy and removed. The final release merges 3- and 4-leg into a single *intersection* class. Figure labeling was aligned to the two-class schema, and outdated phrasing was eliminated.
>
> **Manuscript edits:** Section 3.2 (Data Annotation) clarifies two-class taxonomy; removed stray three-class mentions; caption consistency fixed.
>
> ---
>
> ### 2. Bounding-box tightness variability / missing quantitative agreement
> **Response:**
> We added a paragraph detailing a two-stage annotation + reviewer correction protocol and explicit bounding box guidelines. While we did not compute inter-annotator IoU/F1 statistics for this release, we plan to publish agreement metrics (IoU distribution, class confusion) in the next version.
>
> **Manuscript edits:** Expanded Section 3.2 with protocol paragraph + Appendix (boundary definitions).
>
> ---
>
> ### 3. Loss of structural info by collapsing 3-leg / 4-leg
> **Response:**
> Consolidation avoids severe class sparsity and overfitting. We added justification (Section 3.2) and explicitly flag taxonomy expansion (finer intersection subclasses) in Limitations.
>
> ---
>
> ### 4. Missing deeper analyses (augmentation ablation, seasonal split, failure modes, satellite comparison)
> **Response:**
> Scope constrained to dataset introduction + baseline detection performance. We added K-fold cross-validation (robustness), per-class geometry stats, and a new Related Work paragraph contrasting synthetic datasets. Ablations and systematic seasonal performance tables are reserved for an extended study.
>
> **Manuscript edits:** Sections 4.1, 5.5, 2 (synthetic datasets), 6 (limitations referencing deferred analyses).
>
> ---
>
> ### 5. Potential spatial overlap inflating results
> **Response:**
> Current splits are stratified random; localized geography may produce overlap. We added an explicit statement acknowledging spatial leakage risk and committing to geographic disjoint partitioning (e.g., tile-based splits) in future versions.
>
> **Manuscript edits:** Added to Limitations (Section 6) + K-fold discussion (Section 5.5).
>
> ---
>
> ### 6. Overstated phrases (“comprehensive evaluation”, “outperforming benchmarks”)
> **Response:**
> Wording toned down to emphasize baseline characterization rather than exhaustive superiority claims.
>
> **Manuscript edits:** Abstract, Introduction (Section 1), and Conclusion (Section 7) wording refined.
>
> ---
>
> ## Moderate Issues
>
> ### 1. Per-class distribution and bounding box stats
> **Response:**
> Added class-specific width/height/area/aspect ratio table and narrative analysis (Section 4.1).
>
> ---
>
> ### 2. Bold formatting ambiguity (YOLOv11s)
> **Response:**
> Criteria clarified: selected for best mAP50-95 vs latency trade-off. Changed to plain text without misleading bolding.
>
> ---
>
> ### 3. Weather diversity overstatement
> **Response:**
> Language softened; explicit limitation added (lack of heavy rain, fog, night, active snowfall).
>
> ---
>
> ### 4. Missing synthetic dataset comparison
> **Response:**
> Added paragraph contrasting ROADSIGHT with Syndrone, MidAir, and DDOS in Related Work, emphasizing real-season fidelity and edge benchmarking contributions.
>
> ---
>
> ## Minor Issues
>
> ### 1. Excluding blurry or poorly exposed images (loss of hard negatives)
> **Response:**
> We retained 62 background (no-object) images as hard negatives to mitigate false positives. The curation balances clarity with robustness as noted in Section 3.1.
>
> ---
>
> ### 2. Class imbalance justification reference
> **Response:**
> Added traffic safety distribution citation [1] and directive reference [2] supporting the higher prevalence of traditional intersections.
>
> ---
>
> ### 3. Figure with limited value
> **Response:**
> Removed the figure to streamline the presentation.
>
> ---

---

> ### Author Response · Authors · 2025-12-03
> **Response to Reviewer A4VF (Page 2 of 2)**
>
> ## Responses to Questions
>
> ### Q1: Precise annotation protocol / class merge
> Detailed in Section 3.2: two classes defined for the final release; earlier exploratory three-class drafts were merged prior to quality control. Future extensions will add finer subclasses once adequate sample counts are available.
>
> ---
>
> ### Q2: Geographic independence
> Not fully—localized Finnish regions introduce spatial overlap risk. The Limitations section now states this explicitly and outlines future tile-based geographic partitioning.
>
> ---
>
> ### Q3: Seasonal performance variation
> We provide qualitative seasonal examples; cross-validation stability indicates no catastrophic seasonal overfit. A dedicated per-season evaluation table is planned and noted in Limitations.
>
> ---
>
> ### Q4: Removed low-quality images as hard negatives
> We balanced dataset clarity with inclusion of some clean background images. A future “extended hardness” split including blurred/occluded samples will be released to study robustness under degradation.
>
> ---
>
> ## Closing Statement
> We corrected inconsistencies, enriched protocol transparency, added transformer baselines, per-class geometry analysis, cross-validation robustness, and synthetic dataset positioning. Pending work (geographically disjoint splits, seasonal metric tables, agreement scores, fine-grained taxonomy, adverse weather expansion) is documented in Limitations as concrete next steps.
>
> **References:**
> [1] Zong S, Chen S, Alinizzi M, Labi S. *Leveraging UAV Capabilities for Vehicle Tracking and Collision Risk Assessment at Road Intersections.* Sustainability. 2022; 14(7):4034. https://doi.org/10.3390/su14074034
> [2] Sitran, A., Delhaye, E., & Uccelli, I. (2016). *Directive 2008/96/EC on road infrastructure safety management: an ex-post assessment 5 years after its adoption.* Transportation Research Procedia, 14, 3312–3321.

---

### Official Review · Reviewer_3adf · 2025-10-31

**Soundness:** 2
**Presentation:** 2
**Contribution:** 1
**Rating:** 0
**Confidence:** 4

**Summary:**

The paper proposes a UAV image dataset for detecting road intersections with seasonal variation (winter/summer), annotated as roundabout vs. intersection, and benchmarked with YOLO variants.

**Strengths:**

- Introduces a dataset that is missing from the overall community.

**Weaknesses:**

- This paper reads more like a technical report rather than a research paper.
- The data collection process is quite standard.
- I would expect to see dataset specific analysis. Type of road segments which are more important based on transportation studies
- The motivation is lacking. Why is a UAV necessary to detect a fixed infrastructure that does not change much. Please elaborate to provide a better sense of
- For the type of data and number of images the YOLO performance is not surprising.
- The real-time aspect and edge deployment are not so relevant to the dataset and do not add to the contributions.
- Only YOLO families are reported. For a dataset paper, include at least one non-YOLO baseline (e.g., RT-DETR/DETR-like or a lightweight transformer detector) to avoid family-specific bias and show the dataset’s value across architectures.

**Questions:**

- How did you check for data leakage? Maybe a better way is to do K-fold cross validation and report the average over the folds?

---

> ### Comment · Reviewer_3adf · 2025-11-27
>
> The authors have yet to provide any rebuttal.

---

> ### Author Response · Authors · 2025-12-03
> **Response to Reviewer 3adf**
>
> **We appreciate the candid and critical feedback.**
> Below we group the concerns, indicate manuscript clarifications, and outline concrete future work. Many points are now addressed explicitly in Sections 1, 3, 5, the new Cross-Validation subsection, and the Limitations section.
>
> ## Major Concerns
>
> ### 1. Paper reads like a technical report rather than a research paper
> **Response:**
> We structured the narrative to foreground research questions and explicitly list dataset contributions (end of Section 1), and improved Related Work to position ROADSIGHT vs real and synthetic aerial datasets.
>
> **Manuscript edits:** Introduction (last paragraph), Related Work (synthetic dataset paragraph), Conclusion.
>
> ---
>
> ### 2. Data collection process is standard
> **Response:**
> While UAV capture and bounding boxes are common, ROADSIGHT’s novelty lies in intersection-centric seasonal imagery (winter + summer) at consistent altitudes as well as unified geometry-focused boundary definitions.
>
> **Manuscript edits:** Sections 3.1 (seasonal capture emphasis), 3.2 (boundary protocol).
>
> ---
>
> ### 3. Lack of dataset-specific analysis (e.g., structural relevance of road segments)
> **Response:**
> We added per-class bounding box statistics (Table: class-specific width/height/area/aspect ratios) and discuss how roundabouts exhibit larger footprint distributions, informing scale-aware detection. Transportation safety context retained in Introduction with crash statistics. A direct citation on intersection safety persists [1].
>
> **Manuscript edits:** Section 4.1 (Class and Bounding Box Statistics), Section 6 (Limitation).
>
> ---
>
> ### 4. Motivation for UAV vs. static infrastructure detection
> **Response:**
> Clarified UAV advantages: up-to-date, high-resolution seasonal geometry updates vs. satellite latency; enables rapid deployment for emergency mapping and edge inference without relying on connectivity.
>
> **Manuscript edits:** Motivation paragraph in Introduction.
>
> ---
>
> ### 5. YOLO performance unsurprising for dataset size
> **Response:**
> We acknowledge the moderate difficulty level; to strengthen robustness evidence we added 5-fold cross-validation (Section 5.5) demonstrating stability (low-variance mAP50-95 ~0.669). This addresses concerns about overfitting and generalization.
>
> **Manuscript edits:** New K-fold subsection (Section 5.5) and Table 5.
>
> ---
>
> ### 6. Real-time edge deployment relevance questioned
> **Response:**
> Clarified its practical importance for on-board UAV autonomy (limited bandwidth, latency constraints) and for establishing baselines for researchers targeting embedded decision loops.
>
> **Manuscript edits:** Edge Device Benchmarking section opening paragraph; Limitations note expanded architectures as future work.
>
> ---
>
> ### 7. Only YOLO families reported
> **Response:**
> Added transformer-based baselines (RT-DETR-l, RT-DETR-x) to reduce family-specific bias and situate YOLO trade-offs vs DETR variants.
>
> **Manuscript edits:** Performance table now includes RT-DETR entries (Table 4); Sections 5.2 and 5.3 updated.
>
> ---
>
> ## Responses to Question
>
> ### Q1: Data leakage / preference for K-fold cross-validation
> We performed stratified splits with fixed seed and added a 5-fold cross-validation experiment (Section 5.5), reporting averaged metrics (precision 0.950, mAP50-95 0.669) showing consistent folds and mitigating leakage concerns. Geographic overlap remains a limitation (localized Finnish regions); future releases will enforce spatial partitioning by disjoint geographic tiles.
>
> **Manuscript edits:** Added K-fold subsection (Section 5.5) and limitation clarification (Section 6).
>
> ---
>
> ## Closing Statement
> We added non-YOLO baselines, a formal k-fold cross-validation protocol, and expanded per-class and geometric dataset statistics. Remaining constraints (geographic scope, coarse taxonomy, limited adverse-weather diversity) are transparently stated in Limitations with explicit expansion plans.
>
> **Reference:**
> [1] Zong S, Chen S, Alinizzi M, Labi S. *Leveraging UAV Capabilities for Vehicle Tracking and Collision Risk Assessment at Road Intersections.* Sustainability. 2022; 14(7):4034. https://doi.org/10.3390/su14074034

---

### Official Review · Reviewer_jNjC · 2025-11-01

**Soundness:** 3
**Presentation:** 3
**Contribution:** 2
**Rating:** 4
**Confidence:** 4

**Summary:**

This paper introduces a dataset, ROADSIGHT, which contains of high-resolution UAV-captured images for detecting road intersections in different seasons. The images include annotations of road intersections, categorized into two types: roundabouts and intersections (including 4-leg and 3-leg intersections). The annotations are evaluated on YOLOv11 for edge-optimized performance, ensuring real-time performance on resource-constrained edge devices. This fills a gap in current datasets that detect intersections themselves, and can improve traffic safety and geospatial applications, preventing frequent traffic accidents at intersections.

**Strengths:**

- Current UAV datasets lack intersection detection capabilities; this dataset fills a crucial gap by constructing a high-resolution, multi-seasonal UAV image dataset.
- Considering the robustness and real-time performance issues of models on real-world edge devices, a quantized model is used to enable real-time detection on UAVs.
- The experimental design is rigorous and detailed, with comprehensive evaluation perspectives.

**Weaknesses:**

- The image dataset suffers from homogeneity and exhibits limited detection and processing of complex terrains.
- The paper compares the size differences between the roundabouts and intersections datasets and explains the reasons, but the intersections dataset itself lacks classification information, and its geographical conditions are relatively uniform, raising concerns about dataset diversity.
- The paper focuses on intersection images in summer and winter, but fails to adequately showcase corresponding environmental factors, such as rain and snow conditions.

**Questions:**

- The dataset primarily contains images of intersections with minimal occlusion, but the reality is that intersections often experience significant dynamic occlusion and disturbance. The question remains whether effective annotation can be achieved under these conditions.
- What is the basis for the intersection classification granularity? Does this granularity adequately encompass all types of intersections and ensure good generalization performance of the subsequent recognition model?
- What is the basis for determining the recognition boundary of intersections in the intersection dataset? How are boundary standards unified for different types of intersections?

---

> ### Author Response · Authors · 2025-12-03
> **Response to Reviewer jNjC**
>
> **We thank the reviewer for the thoughtful and constructive feedback.**
> Below we provide concise responses to each concern and summarize the corresponding manuscript revisions.
>
> ## Major Concerns
>
> ### 1. Geographic Homogeneity
> **Response:**
> We acknowledge the limited geographic scope (localized Finnish regions). This is now clearly stated as a limitation in Section 6. Our initial goal was to build a controlled, high-quality, seasonally varied dataset. To mitigate homogeneity during training, we employed stratified splits and 5-fold cross-validation (Section 5.5, Table 5), as well as targeted augmentations (Appendix). Future expansion plans (additional terrains and regions) are now explicitly noted in Section 6.
>
> **Manuscript edits:** Added limitation statement; added cross-validation analysis.
>
> ### 2. Lack of Finer Intersection Classes
> **Response:**
> We intentionally restricted classes to *roundabout* and *intersection* to maintain sufficient per-class sample sizes and avoid overfitting. Sub-classes (T-, Y-, 3-/4-leg) were avoided due to undersampling. This design rationale is now clarified in Section 3.2 and Section 6. We plan to introduce finer taxonomy once sample counts increase.
>
> **Manuscript edits:** Added explanation in Section 3.2 and limitation note in Section 6.
>
> ### 3. Limited Weather Diversity
> **Response:**
> The dataset includes varying levels of snow and lighting, but lacks heavy rain or dense fog. This is now stated as a limitation in Section 6. Robustness was partially addressed through photometric augmentations and by reporting stable results across cross-validation folds (Section 5.5).
>
> **Manuscript edits:** Updated Section 6 accordingly.
>
> ## Responses to Questions
>
> ### Q1: Annotation under dynamic occlusion
> We agree this is an important challenge. This release focuses on intersection geometry only, without per-object annotations, to maintain GDPR compliance and annotation consistency. Future versions will include occlusion metadata and object annotations once anonymization workflows are established.
>
> **Manuscript edits:** Added explanation in Section 3.2.
>
> ### Q2: Basis for classification granularity
> The two-class scheme maximizes sample balance and generalization for the current dataset size. K-fold validation (Table 5) demonstrates stable model performance under this granularity. Clarified in Section 3.2.
>
> ### Q3: Unified boundary definitions
> A clear, reproducible boundary-definition standard—based on civil-engineering literature—has been added.
>
> - **Intersection:** conflict area + approach legs + auxiliary elements.
> - **Roundabout:** outer circulatory roadway + central island + splitter islands (+ truck apron).
>
> Referenced sources are included in Appendix A.
>
> **Manuscript edits:** Added to Section 3.2 and Appendix.
>
> ## Closing Statement
> **ROADSIGHT** is positioned as a curated, seasonally diverse UAV dataset focused on intersection geometry and edge-optimized detection. We have clarified all limitations and documented our plans for dataset expansion (more diverse regions, weather conditions, occlusion metadata, and finer taxonomies).

---

### Meta-Review · Area_Chair_Vw8z · 2026-01-07

**Summary:**

The study introduces ROADSIGHT, a novel dataset specifically designed for real-time intersection detection in unmanned aerial vehicle (UAV) imagery, notably addressing the crucial need for seasonal variation in aerial computer vision benchmarks. The paper establishes robust performance metrics using YOLO architectures and successfully demonstrates the practical viability of deploying an optimized model, YOLOv11s, on resource-constrained edge hardware, achieving real-time performance of 69 FPS on the NVIDIA Jetson Orin Nano. This work addresses limitations in prior datasets, which often lack intersection specificity or environmental diversity.

Reviewers expressed concerns including:

(1) the dataset scale is relatively modest and geographically limited. Containing 969 total images collected only from localized test areas in Finland, the limited geographic diversity may hinder the model’s generalization capacity when applied to regions with significantly different road infrastructures or climates.
(2) The granularity of intersection classification is insufficient for detailed analysis. Although the initial collection included roundabouts, 3-leg (T/Y), and 4-leg intersections, the final annotations simplify these categories into only two: “roundabout” and a merged “intersection” class (for T/Y and 4-leg types). This limited classification may restrict the dataset’s value for applications requiring distinctions among specific intersection geometries. Further, the classification on finer classes is not evaluated.
(3) Loss function weights are not justified empirically or theoretically, nor are model decisions.
(4) No comparison of model accuracy on non-curated data.
(5) More severe cases of weather are not included in the dataset (e.g. heavy fog).
(6) Annotation guide or analysis would be important to establish reliability of annotations.
(7) The same intersection may appear in multiple splits.
(8) Clarification on micro vs macro averaging of per-class performance is missing.
(9) Formatting and reference suggestions.
(10) The dataset primarily contains images of intersections with minimal occlusion, but the reality is that intersections often experience significant dynamic occlusion and disturbance. The question remains whether effective annotation can be achieved under these conditions.

**Reviewer Concerns:**

(1-5) are outstanding, as the dataset scale and granularity is unchanged, loss weights are online tool defaults, and no comparison to non-curated data is performed.

(6, 8, 9) have been added or improved to increase strength of the paper.

(7) has been acknowledged, but the reviewer's observation is correct (the same intersection may appear in multiple splits), which is a weakness of the paper.

Authors reply that (10) is left for future work.

**Reviewer Scores:**

KWuN: unchanged (6). Comments are replied to, but none of the perceived weaknesses are strengthened in argument or revision.
A4VF: unchanged (2). Data is not split geographically for validation.
3adf: unchanged (0). Concerns around data splits and model choice are similar to other reviewers.
jNjC: unchanged (4). Concerns about data quality / features remain unaddressed.

---

### Decision · Program_Chairs · 2026-01-26

Reject